# An Adaptive RF Front-End Architecture for Multi-Band SDR in Avionics

**DOI:** 10.3390/s24185963

**Published:** 2024-09-14

**Authors:** Behnam Shakibafar, Farzan Farhangian, Jean-Marc Gagne, Rene Jr. Landry, Frederic Nabki

**Affiliations:** LASSENA Laboratory, Department of Electrical Engineering, École de Technologie Supérieure, Montreal, QC H3C-1K3, Canada; behnam.shakibafar@lassena.etsmtl.ca (B.S.); farzan.farhangian@lassena.etsmtl.ca (F.F.); jean-marc.gagne@lassena.etsmtl.ca (J.-M.G.); renejr.landry@etsmtl.ca (R.J.L.)

**Keywords:** avionics, software defined receiver, SDR, front-end

## Abstract

This study introduces a reconfigurable and agile RF front-end (RFFE) architecture that significantly enhances the performance of software-defined radios (SDRs) by seamlessly adjusting to varying signal requirements, frequencies, and protocols. This flexibility greatly enhances spectrum utilization, signal integrity, and overall system efficiency—critical factors in aviation, where reliable communication, navigation, and surveillance systems are vital for safety. A versatile RF front-end is thus indispensable, enhancing connectivity and safety standards. We explore the integration of this flexible RF front-end in SDRs, focusing on the detailed design of essential components, such as receivers, transmitters, RF switches, combiners, and splitters, and their corresponding RF pathways. Comprehensive performance evaluations confirm the architecture’s reliability and functionality, including an extensive analysis of receiver gain, linearity, and two-tone test results. These assessments validate the architecture’s suitability for aviation radios and address considerations of size, weight, and power-cost (SWaP-C), demonstrating significant gains in operational efficiency and cost-effectiveness. The introduction of the new RF front-end on a single SDR board not only substantially reduces size and weight but also adds up to 18 dB gain to the received signal. It also allows for a high level of design flexibility, enabling seamless software transitions between different radios and the capacity to manage three times more radios with the same hardware, thereby significantly boosting the system’s ability to handle multiple radio channels efficiently.

## 1. Introduction

Avionics systems rely on robust and flexible communication, navigation, and surveillance (CNS) capabilities. A critical component of these systems is the RF front-end (RFFE), which interfaces with antennas, receivers, and transmitters to enable wireless communication. Software-defined radios (SDRs) play a crucial role in advancing RF and wireless communication technologies, enhancing flexibility and performance metrics. They are particularly utilized in various technologies, such as integrated modular avionics (IMA) and satellite communication systems for high-throughput data links in various orbits. These systems employ a flexible radio architecture with baseband algorithms implemented in field-programmable gate arrays (FPGAs) or digital signal processors (DSP). SDRs offer the advantage of modularity, expandability, updating processing algorithms in space, reducing costs, and minimizing size by integrating most processing functions into a single chip. Despite their modular design, current SDRs are often tailored to specific applications due to unique RF and analog–digital conversion requirements.

Efforts to enhance the size, weight, and power–cost (SWaP-C) efficiency of RF avionics in recent decades involved the adoption of digital avionics and IMAs. However, these approaches faced limitations within existing RF architectures, particularly in terms of accommodating simultaneous applications on a single digital avionics front-end platform. Conventional RF architectures posed constraints on various industries, including aviation. The emergence of higher-resolution, higher-speed analog–digital converter (ADC) or digital-to-analog converter (DAC) technology allowed for the introduction of the DRFS architecture as an alternative design. The limitations of the RFFE design adversely affected the accuracy and performance of various air navigation systems, such as very high frequency omnirange (VHF), automatic dependent surveillance broadcast (ADS-B) system, distance measuring equipment (DME), tactical air navigation (TACAN), and VHF omnidirectional range (VOR). Apart from air navigation technologies, RFFEs are widely used in designed SDRs for radio communication (COM); transponder mode-s (TMS); and different instrument landing systems (ILS), such as instrument landing system localizers and glide slopes (ILS LOC and ILS GS). Table 1 shows the comparison of different radio avionics based on the frequency, bandwidth, modulation type, and receiver/transmitter (RX/TX) power levels.

Size, weight, and modularity are important factors for avionic radios. Using multiple avionic air navigation systems in a single test needs an integrated and reliable front-end to be flexible enough to satisfy the requirements of different radios. In the proposed work, we present an agile RFFE architecture designed for SDRs in avionics applications. The approach in this paper focuses on establishing multiple RF paths shared among radio modules with the same technical requirements to reach a size and weight optimization. Other principal components of the design have been designed to keep the flexibility and add ease of debugging for integrated applications in which power consumption and system integrity are the vital criteria. Moving forward, an essential aspect of this work is ensuring the readiness of the designed architecture with minimal configuration for flight tests, the next critical step in this research. Factors such as antenna placement, external cabling outside of the RFFE and safe equipment adjustment will all be considered as the RFFE is prepared for flight testing. This work is primarily intended to present the design, including port placement, size and weight optimization, and also the mobility of the design for light aircraft, where cable losses are minimal and temperature variations are generally consistent with room temperature conditions.

In this paper, the proposed agile RFFE design is presented in the following sections. The related literature of the previous efforts made in this area is discussed and presented in Section 2, followed by a discussion of architecture design in Section 3. This section aims to introduce a new integrated methodology for a single front-end module. Section 4 demonstrates the integration technique to make the design practical for the flight test. Then, Section 5 and Section 6 concentrate on the performance evaluation of the proposed method in lab experiments. Various performance metrics are introduced and investigated in this section to show the efficiency of the design. Finally, a brief discussion and conclusion are given in Section 6 and Section 7, respectively.

## 2. Background

RFFEs have been extensively researched and applied for a variety of applications. As small satellite platforms require smaller SWaP-Cs, the Johns Hopkins University Applied Physics Laboratory has designed a single-board radio tailored for small satellite architectures to enhance its proven Frontier Radio (FR) system. A dual-receiver RF card with a flexible transmit IF allows four traditional Frontier Radio slices to be combined on one RF card [1]. The authors in [2] examined the benefits of modular, open, integrated architectures in military and commercial avionics, which offer a cost-effective approach to addressing dynamic technological changes. In this particular case study, the avionics architecture was optimized to meet evolving technology and mission requirements for an aircraft system through scalability in both hardware and software. Open-source hardware and software have been released to focus on the communication modules of IMAs for remote piloted aircraft systems. There has been some effort made in [3] to make communication links in integrated modules secure, in order to simplify certification for these avionic architectures. 

SDRs have shown many benefits not just in terrestrial RF applications but also in the space industry, which traditionally relied on conservative approaches. The development of programmable and reconfigurable RF integrated circuits (RFICs) has opened up new possibilities for improving flexibility. A generic software-defined radio (GSDR) platform capable of dealing with space-specific environmental conditions was discussed in [4]. Other applications, including civil aviation and air transportation, also rely on enhanced front-end platforms. It was the direct RF sampling (DRFS) architecture that indicated a proper response to these escalating demands [5], specifically designed for small and private aircraft, narrow bodies and wide bodies, drones, etc. Radio communication and navigation systems are crucial to the safety of civil airplanes during flight.

A single integrated front-end for all avionic modules has been investigated in research works. For instance, a unified hardware communication and navigation ground simulation system was developed utilizing SDR technology and an inexpensive and versatile hardware platform [6]. Various tests and performance analyses performed in the lab environment proved the feasibility of an integrated design to accommodate multiple airplane systems with varying operating frequencies (HF, VHF, VOR, ILS, and ADF). Another example is the aviation microwave-integrated circuit (AVMIC), which is designed to reduce the weight of physical circuits, increasing range, fuel efficiency and passenger capacity [7]. Similarly, a performance analysis of wideband radio (WBR) for a software defined avionic module (SDAM) was published in response to the increasing number of passengers. During the flight, they analyzed the transmission power and its correlation with the maximum slant range achievable by module [8].

In our prior work, we have attempted to improve RF avionics security and performance by addressing its critical requirements. A DRFS architecture was used to limit the number of RF components between the antenna and receiver, supporting multiple avionics applications simultaneously [9]. This led to many flight tests, covering scenarios, installation, configuration, and performance analysis. This study confirms the multi-mode SDR’s operational effectiveness in flight conditions, providing valuable insight for avionics architecture development and outlining its advantages and limitations for future advancements [10].

VOR, ILS, DME, and other avionics RF functionalities must meet SWaP-C requirements, especially with the proliferation of unmanned aerial systems. By using a DRFS approach, the work in [11] explored new hardware implementations of the digital downconverter architecture on FPGAs. The development of integrated avionics has not stopped here, and various papers have investigated multiple-input/multiple-output (MIMO) architectures to achieve a modular structure in SDRs. By using pre-selection filters and FPGAs, the researchers in [12] proposed an architecture combining a digital receiver with SDR software modules. This can be utilized for different applications by utilizing a variety of modulation modes and RF/IF frequency bands. That design could eventually surpass equivalent commercial systems in terms of miniaturization, lightweight design, and low power consumption [12].

An innovative concept called “complex domain” RF was introduced in [13] to address the challenges associated with phased array-based digital beamforming systems. By using a technique that separates waveform delay information for adaptive beamforming from wideband RF signals, a self-contained beamforming system can be implemented with a low-speed baseband [13]. One of the most efficient SDRs used in the avionics industry is the Xilinx RFSoC, which was used in [14] to design a software-defined radar altimeter. Multiple experiments with certified equipment (Alt-8000) demonstrate that the proposed architecture follows the accuracy standards outlined in RTCA’s DO-155 [14]. Other work presented a 24 GHz RF TX front-end for radar applications by integrating an up-conversion mixer and power amplifier (PA) [15]. The proposed TX front-end could achieve an output power of 11.7 dBm, dissipation of 7.5 mW, power-added efficiency (PAE) of 47%, and 1 dB compression point (OP1dB) of 10.5 dBm. This design holds promise for advancing 24 GHz radar technology, with potential applications in autonomous vehicles, industrial automation, and remote sensing.

The integration of SDRs within space communications architectures is pivotal, enhancing the framework of next-generation integrated communication systems. This integration fosters reconfigurability and builds upon insights from early SDR applications, such as those seen in NASA’s Space Communications and Navigation (SCaN) programs [16]. Additionally, the research in [17] investigated enhancements in avionics navigation solutions through the integration of air navigation technologies, specifically focusing on alternative positioning, navigation, and timing (APNT) systems like DME and very high frequency omni-directional range (VOR). This study also explored pulse pair design compliant with the DO-178B avionics standard, evaluating various guidelines and potential for certification. Laboratory tests using an IFR6000 confirmed that the proposed architecture could achieve a positional accuracy of less than 0.23 nautical miles with a 98% confidence level [17].

## 3. System Architecture

The proposed agile RFFE design combines multiple radios into a single structure, connecting them to an RFSoC4x2 (i.e., SDR). This setup offers key benefits: flexibility to add or upgrade components easily, simplified debugging, and effortless preparation for lab or flight tests. This section first outlines each block depicted in Figure 1, detailing its components, intended outcomes, and design process in detail. Then, simulations of the receiver blocks are presented.

### 3.1. Design Blocks

The architecture comprises essential design blocks: receivers, transmitters, RF switches, splitters, and combiners. Each block plays an important crucial role in the RFFE.

Receivers: The receiver block includes a low-noise amplifier (LNA), bandpass filter (BPF), and RF limiter. These components enhance signal sensitivity and selectivity, and they protect against high-power interference.Transmitters: The transmitter block integrates an off-the-shelf high-power amplifier (HPA) and input BPF. This ensures efficient transmission, without unwanted signals conducted to the input of the amplifier.RF switch, splitter, and combiner: These elements are used to control connections to the antenna and RF pins of the RFSoC, enabling seamless switching between radios and RF IOs.

### 3.2. RF Paths

RF paths within the architecture utilize the design blocks mentioned above. Figure 1 provides an overview of how these blocks and paths are interconnected. Each path uses a set of blocks to match the requirements of a certain radio, including the center frequency, bandwidth, sensitivity, and output power. Due to the similarity in center frequency and bandwidth requirements for some radios, we have implemented a strategy where the same RF path is shared among them. An example of this can be observed in the block diagram, where the VOR and ILS LOC radios share the RX path alongside the TMS, DME, and ADS-B radios, which, in turn, share both RX and TX paths. The software seamlessly manages the transition from one radio to another, leveraging the same RF RX and RF TX pins on the RFSoC.

Another noteworthy aspect is the use of the same RX pin for multiple RX or TX paths. As depicted, the VOR, LOC, and GS radios are all connected to the same RF input across two different RF paths. These two paths maintain a sufficient frequency gap, allowing for straightforward combining and subsequent distinction through both hardware and software. Similarly, the COM and TMS/DME/ADS-B paths share the same RF output, following a similar approach of combining and distinguishing the signals within the software, owing to the availability of adequate frequency space.

There are also multiple GPIO pins used to control the RF switch and power amplifiers to make sure that the software has the full control needed to utilize RX and TX over the same antenna, heat management, and safety, which are detailed in Section 4.4 and Section 4.5.

### 3.3. Receiver Block Simulation

The main purpose of RFFE at the receiver side is to improve system sensitivity by adding gain to received signal. The same as any other RFFE, the SDR receiver has the responsibility of filtering and amplifying the signal that is received by the antenna. Figure 2 shows the block diagram of the receiver blocks. Each has three main components: the limiter, which protects the rest of the RF path against high power signals (over 12 dBm); the amplifier, which amplifies the signal; and, lastly, the bandpass filter, which is filtering out all of the band signals. This design is similar for all receive paths, except the bandpass filter, which should match the center frequency and bandwidth of the signal of interest.

Before proceeding with implementation, the gain of the receiver is simulated using PathWave RF Synthesis (Genesys). Figure 2 depicts the block diagram of the simulated RF path, where the first and second components (RF limiter and low-noise amplifier) remain consistent across all receiver path radios, while the third component (bandpass filter) is adjusted to match the frequency bandwidth of the target radio. To ensure precision, all models utilize manufacturer-provided S-parameter models for an environmental temperature of 25 °C. Figure 3, Figure 4, Figure 5 and Figure 6 illustrate the simulated gain (S21) and input matching (S11) for the L-band receivers (including TMS, DME, and ADS-B), ILS Glide Slope, VOR/ILS Localizer, and VHF communication, respectively. The observed gain for all radios in the frequency bands of interest are consistently over 19 dB, and input matching is achieved, indicating that the selected components and the overall design path align well with the initial expectations set during the design phase.

Table 2 presents the RF budgeting for the receiver block. Given that, except for the bandwidth and frequency, the BPF specifications are nearly identical for all radios, this table is applicable to other designs as well. The receiver is capable of delivering up to 1.3 dBm to the RFSoC pin, comfortably covering the full measurement range of 1 dBm and remaining well within the safe range for the ADC input, which is 14.6 dBm. Additionally, the RFSoC has an internal tunable attenuator, so we are not concerned about distortion and clipping of the signal.

The integration process focuses on configuring the architecture for seamless deployment in both lab and flight test environments. The system’s key requirements include easy installation, portability, hot-swapability, and resilience to vibrations and various flight conditions. To accomplish these objectives, two dedicated rack units have been developed for L-band and VHF purposes. Each unit is designed to utilize a shared 28 V power supply.

## 4. System Design and Integration

This section provides a comprehensive overview of the implemented design, including the RF pathway and its associated components, the power supply mechanisms, and considerations for thermal and spatial management. It delves into the details of how signals are propagated through the RF path and how critical factors such as power management, heat dissipation, and space constraints are addressed to optimize the design.

To achieve optimal performance, it was crucial to select and design each component with precise specifications, prioritizing efficiency in size, weight, and power usage. Among the most critical components is the receiver block, which not only handles various voltage levels through DC-DC converters but also manages the interfacing for input and output RF signals using standard SMA connectors. This section also details the structural design and provides a description and visualization of the ports.

### 4.1. Design Flexibility

As shown in Figure 7 and Figure 8, each box consists of 12 SMA positions, which allow the design to maintain flexibility for adding new inputs and outputs, accommodating modifications to radios and their connections. At each box, the control signal supports up to 12 signals; however, only two are utilized in this work—one for PA shutdown and the other for RF switch control. The 28VDC power supply serves as the main bus due to its common use in many PAs. Additionally, switching DC-DC converters are employed to achieve various voltage levels, specifically 5 V and 12 V, as demonstrated in this project.

The L-band box supports all radios in the bandwidth range from 840 MHz to 1440 MHz with up to 100 W output power, while the VHF box can handle up to six radios with narrower and interchangeable BPFs in the receiver blocks.

This design also supports adding new boxes, including the processor box with the RFSoC 4 × 2 board. All boxes are designed to fit perfectly into a four-stage container, facilitating ease of mobility and flight-testing procedures.

### 4.2. VHF Radio Design

One radio rack unit is dedicated to consolidating the VHF RF paths to ensure reliability during flight tests. This rack supports VOR, ILS LOC, ILS GS, and VHF COM functions. Specifically, the VOR and ILS LOC radios share an RX RFFE, while the ILS GS and VHF COM have dedicated RX and TX front-ends. The receiver block uses the same design as described in Section 3.3. It includes an RF limiter (RLM-33+), followed by an LNA (GALI-S66+) and a bandpass filter tailored to match the target bandwidth. Specifically, the BPF-A127 is used for COM, the BPF-A113 is used for VOR and ILS LOC, and the BPF-A332 is used for ILS GS. For the rest of the unit, some off-the-shelf RF components were utilized to mitigate risk, ensure robust operation, and allow for relatively short design time. A 20 W power amplifier (Ophir 5303027) is employed to ensure sufficient gain and output power, while an RF switch facilitates seamless switching between RX and TX modes during operation. Given that the switching speed is not excessively rapid (i.e., less than 100 ms), the primary consideration for selecting the appropriate RF switch was its maximum power-handling capacity. After evaluating all requirements, a mechanical RF switch (Mini-Circuits MSP2T-18-12+) was chosen and subsequently tested.

As mentioned earlier, two DC-DC converters (PQDE6W-Q48-S12-T) are used to con-vert a 28 V DC input to a 12 V DC supply for the receiver and RF-switch modules. Since the power amplifier operates on a 28 V DC supply, no DC-DC converter is needed to power it. On the other hand, the LNA itself operates from 3 V to 4 V. The circuit design used a resistor value suggested by the datasheet that provides the proper bias current at the 12 V DC supply. Figure 7 shows the rack unit for the VHF radio and the description of its ports, respectively.

### 4.3. L-Band Radio Design

Another radio rack unit is specifically designed for L-band RF pathways to support ADS-B, DME, and TMS functions. All three radios in this band share the same frequency range, enabling the integration of a unified receiver and transmitter in the RFFE. The RFFE in this rack incorporates essential components such as RF protection and a low-noise amplifier. It also features a bandpass filter for the receiver and another bandpass filter and a power amplifier for the transmitter. An additional element included is a variable attenuator, which is controlled by a power detector; it is used to finely adjust the RX pin signal levels on the RFSoC.

At the receiver side, the same design used as the VHF unit was used, but with a suitable BPF (BPF-A1140). The same DC-DC converter (PQDE6W-Q48-S12-T) was used to convert the 28 V DC to a 12 V DC supply for the receiver block. The same as the VHF unit, the receiver block has a proper series resistor, which provides the needed bias current at the 12 V DC supply. The power amplifier in this system is tasked with delivering up to 47 dBm (50 W) of power at the output, utilizing a class AB amplifier. These amplifiers typically operate with an efficiency ranging from 50% to 70%. Consequently, managing heat radiation becomes crucial due to the dissipated power. The most crucial part of this system is the RF switch. During the design phase, we prioritized finding one that could handle maximum power (at least 100 W) and switch quickly (within 4 µs). To meet these needs, we opted for a GaN-based RF switch known for its efficiency and speed (SKY12245). A development board from the manufacturer was used in our design for this RF switch (SKY12245-492LF). The module is powered using a DC-DC converter (PQDE6W-Q24-S5-T) that generates 5 V DC from the 28 V DC input. Additional information on maximum power-handling schemes, test procedures, and other considerations can be found in Section 4.4. Figure 8 shows the L-band rack unit implementation and a description of its ports, respectively.

To ensure temperature stability and good heat dissipation of the power amplifier, eight fans are installed on the back of the panel. These fans are powered by a 10 W DC-DC converter (PYB10-Q48-S12-T), bringing the 28 V DC supply to 12 V DC.

### 4.4. RX/TX Mode Handling

In this system, like any other transceiver, certain radios necessitate the ability to both transmit and receive using a single antenna. Operating as an SDR, the software is tasked with managing the seamless transition between these two modes, while the RFFE facilitates a safe and reliable switch. This critical role is fulfilled by the RF switch.

As depicted in Figure 1’s block diagram, both the L-band and VHF band modules incorporate an RF switch. Since the VHF band COM operates as a half-duplex radio controlled by a manual switch with a very low speed transition-time sensitivity (around 100 ms), switching between RX and TX modes is not a significant challenge and can be readily managed by a control signal.

In the case of the L-band module shown in Figure 9, the power amplifier can output power levels as high as 47 dBm (50 W), while the receiver block endeavors to amplify signals as faint as −90 dBm. The RF switch must adeptly navigate between these two extremes, transitioning from RX mode to TX mode when the SDR is transmitting, and reverting to RX mode to listen to the band.

Figure 9 illustrates how the RF switch is deployed to conduct the PA output to the antenna, effectively managing the transmission and reception processes within the system. To enhance system performance, potential improvements could involve optimizing the RF switch for faster transition times between modes, minimizing signal loss during switching, and ensuring robustness to handle varying power levels and frequencies encountered in real-world scenarios. Additionally, enhancing the software’s control algorithms could lead to more efficient and seamless handoffs between transmission and reception modes, minimizing any damage to the RF switch and the RX path.

Table 3 presents the maximum power deliverable to the ADC input of the RFSoC in TX mode. The typical loss or gain, along with the maximum possible power at each stage, is detailed. The total potential leakage power to the receiver is measured up to 3.3 dBm. This power can be amplified to a maximum of 12.1 dBm by the saturated LNA, which remains below the 14.6 dBm maximum allowable power for the RFSoC’s ADC pin.

Selecting an appropriate switch capable of handling 50 W in the main harmonics and up to 100 W total power, with high isolation levels and a swift switching time of less than 4 µs, is important for the design. The Skyworth SKY12245 RF switch emerged as the optimal choice. This switch achieves approximately 45 dB isolation, meaning the 47.75 dBm output (equivalent to 59.56 W) results in a minor leakage of about 3.3 dBm or 2 mW into the RX path. This leakage is considered negligible and raises no significant reliability concerns. The test setup for assessing the timing and isolation of SKY12245 is shown in Figure 9. A dashed line is used to represent the control signal, and a 50 dB attenuation is employed to protect the spectrum analyzer. It is implemented by a 30 dB 200 W SA3N200-30 by a Fairview microwave cascaded with a 20 dB 500 mW CATTEN-0200 by Crystek Corporation. Figure 10a presents the highly effective timing scheme. The green signal activates the power amplifier while simultaneously setting the RF switch to TX mode, ensuring there is no risk of a high-power signal being conducted to the RF switch in RX mode. A 4 µs time slot is allocated to cover the switching transition time before the commencement of signal transmission (orange signal) by the SDR. The frequency-domain signal illustrates the transmitted-signal power spectrum, validating the signal shape and power. Considering the 50 dB attenuation, the measured signal strength of −2.25 dBm indicates a transmitted signal of 47.75 dBm to the antenna port.

Figure 10b also illustrates the impact of the signal transmission on the RX path during TX operation, confirming the approximately 45 dB TX to RX isolation claimed in the datasheet of the SKY12245 switch. Both the yellow signal in the time domain and the orange spectrum confirm a very small leakage signal, measuring at −46 dBm. Considering the 50 dB attenuation used in the measurement, the actual signal strength is +4 dBm at the receiver, which is as expected considering the switch isolation specification of approximately 45 dB and the transmitted signal of 47.75 dBm.

### 4.5. Heat Management 

One notable enhancement in the Swap-C project was the significant reduction in heat generation by the PA. This breakthrough enabled the final enclosure to be considerably smaller and lighter, eliminating the need for an additional bulky heatsink. While transmissions in aviation communications are not continuous, having good time division can help to reduce heat generation further.

DME operates with signal bursts consisting of two 3.5-microsecond pulses separated by 12 μs, totaling about 19 dBs per burst. These bursts are sent every 21.3 ms for Modes A and C, and every 12.5 ms for Mode S, using a bandwidth of about 1 MHz. Similarly, ADS-B transmits 120-microsecond messages every 0.5-to-1 s in the 1090 MHz band, also occupying around 1 MHz bandwidth. TMS, used for secondary surveillance radar, transmits data in discrete bursts of 56 or 112 μs, depending on the message type, and operates in the 1090 MHz band. Due to the high-frequency transmission and significant bandwidth usage, these systems generate substantial heat. By shutting down the PA between transmissions, which is a major source of heat during these transmissions, thermal control can be achieved, preventing overheating and maintaining optimal system performance. This was accomplished by implementing a control mechanism for the PA, allowing it to enter shutdown mode when no signal transmission was occurring. Through this approach, the SDR activates the PA just 4 microseconds prior to transmission, as per the PA datasheet specifications. This brief activation window ensures efficient signal transmission, while minimizing heat buildup. Once transmission is complete, the PA seamlessly returns to shutdown mode, further enhancing energy efficiency and reducing overall thermal output. This optimization not only enhances the performance of the Swap-C system but also contributes to its portability and versatility in various operational environments. The green signal in Figure 10 shows how the process was handled, led by the software. Figure 11a shows the probes’ placement, and Figure 11b shows the test setup used for long-term temperature measurement. As can be seen, two points are probed with the thermocouple: the body of the LNA enclosure and the body of the PA enclosure. The long-term temperature results are reported in Table 4, where the highest recorded temperature on the PA body is reported to be 47.1 °C after two hours of continuous transmit operation, as described above. The test continued for an additional 30 min to ensure that the temperature was stabilized at around 47 ± 1 °C.

## 5. RF Performance

Measuring the gain and linearity in RF receivers and transmitters is crucial for ensuring the stability and reliability of communication systems. Gain measurement is key for providing effective signal amplification, avoiding excessive noise, and ensuring long-distance signal transmission. Linearity measurement is vital for accurate signal reproduction without distortion, preserving the integrity of transmitted information. The proper calibration and testing of these parameters are essential to prevent signal degradation, interference, and reliability issues, thereby avoiding potential communication failures. Rigorous testing and precise measurements are required to ensure that systems meet specified performance standards. This section details how the RF performance was evaluated.

### 5.1. Receiver Noise Figure

The noise figure is an important parameter in RF design that can directly impact the sensitivity of the receiver. As shown in Figure 12, the noise figure was measured and found to be between 1.2 dB and 1.4 dB within the bandwidth of interest. The peak of 23 dB in the noise figure at 875 MHz, which corresponds to the LTE band, could be due to interference from nearby LTE signals in the laboratory. This interference can raise the noise floor, resulting in the observed increase in the noise figure at this frequency. Moreover, the measured in-band gain matches well with expectations.

### 5.2. Receiver Gain

According to the datasheet specifications of the components utilized, the overall amplification of the system is expected to range from 19 dB to 21 dB across various receiver paths. Table 5 details the gain and insertion loss for the components used in each radio receiver block. 

Both the measured output signal and gain for each radio are reported in Figure 13. The measurements indicate a gain of between 17 and 20 dB for all radios, with input signal levels ranging from −90 dBm to −20 dBm. Additionally, the signal level remains nearly consistent across all frequencies at each input level, suggesting good linearity in gain. This consistency is further supported by the use of the same circuit design, with careful component selection for all radios, while only adjusting the bandpass filter (BPF) for the target bandwidth. This approach simplifies both the design and implementation processes.

The output power and gain demonstrate linearity up to the 1 dB compression point. The values for IP3 and OIP3 of each receiver are outlined in the next measurement. However, output-signal levels must be kept under 7 dBm to protect the SDR’s ADC input. 

### 5.3. Two-Tone Test

The test configuration is modified to accommodate the need for generating signals with closely spaced frequencies. To achieve this, two signal generators are employed and are fed into a combiner, which in turn supplies the input signal to the device under test (DUT). The output from the DUT is observed using a spectrum analyzer, as shown in Figure 14.

The current configuration has its drawbacks, primarily due to the power splitter acting as a combiner, which introduces some power loss in the signal path. However, this was required since our signal generator did not include a two-tone mode.

In lab experiments, the 1 dB compression (OP1dB) and third-order input intercept IIP3 for each frequency band were measured, and they are reported in the following figures. Starting with VOR, ILS, and COM, Figure 15 shows the experiment results for a two-tone test at 108 MHz and 110 MHz

The results show an OP1dB of 2 dBm, an Output IP3 (OIP3) of 28 dBm, and an Input IP3 (IIP3) of 18 dBm.

Figure 16 shows the OP1dB and IIP3 measurement results for the ILS Glide Slope at a 330 MHz center frequency and 332 MHz for second frequency. As can be seen, the OP1 is 3 dBm, while the OIP3 and IIP3 are 27 dBm and 16 dBm, respectively.

The last two-tone test was performed at 1090 MHz and 1092 MHz, which are the operating frequencies for DME, ADS-B, and TMS. Figure 17 shows measurement results. The figure shows an OP1dB of 2 dBm, OIP3 of 10 dBm, and IIP3 of 1 dBm.

As an additional test, the input signal power of all components was recorded at each input level. The output spectrum reveals the fundamental frequencies at 1090 MHz, along with second-order harmonics at 2180 MHz (2 × 1090) and 2182 MHz (1090 + 1092), and third-order intermodulation distortion (IMD) products at 1094 MHz (1090 + 1094 − 1090). These harmonics and intermodulation products indicate the nonlinearities present in the LNA, which are critical for evaluating its suitability for applications requiring high linearity and low distortion. The presence and levels of these harmonics help in assessing the amplifier’s ability to amplify signals without significant distortion, which is essential for maintaining signal integrity in communication systems. This demonstrated a significant margin between the fundamental and other components before reaching a 1 dB compression point. The results are illustrated in Figure 18.

In summary, although we observed degradation in LNA performance at higher frequencies, both IP1 and IP3 values are within satisfactory ranges, aligning with system specifications. Furthermore, Figure 18 illustrates good ratios in power between all harmonics. These results reaffirm the overall robustness of each receive path’s linearity performance, guaranteeing reliable signal reception across the frequency spectrum targeted by radios in the system.

## 6. Discussion

The primary objective of this work was to seamlessly integrate various radios within the RFFE of the SDAM while concurrently reducing SWaP-C considerations and fulfilling a vast array of operational requirements. The project achieved the implementation of seven distinct radios within a relatively compact physical footprint, constituting less than half the area occupied by the previous iteration [8,10,11]. Through meticulous design and testing protocols, the system passes target radio requirements.

The advantages of the proposed architecture are multifaceted, demonstrating significant advancements in several key areas:System Size Reduction: The integration of multiple radios into the same RFFE has achieved a substantial reduction in total size, weight, and cost. This efficiency gain is critical in contexts where space is at a premium.Enhanced Flexibility: The architecture’s software provides extensive control, enhancing the RFFE’s adaptability for future updates and technological shifts, also contributing to the system’s longevity and relevance in evolving operational contexts.Modular Design: The system’s modular framework supports straightforward upgrades and modifications, facilitating quick adaptations as technology or operational requirements evolve.Reliability and Resilience: Independent functionality for each component minimizes the risk of failure at any single point, thereby boosting the overall system reliability and resilience.Improved Signal Integrity: Enhancements in both transmitting and receiving signal quality promise significant benefits in SWaP-C and adaptability, contributing to more robust and reliable communications.

Additionally, the ability of the system to handle multiple radio channels efficiently enhances its operational capacity, significantly exceeding the rigorous demands placed on contemporary aviation communication systems. These improvements ensure that the system not only meets but also anticipates future requirements, establishing a significant benchmark for avionic RF technologies.

## 7. Conclusions

This paper detailed the development of an agile RFFE system specifically designed for multi-SDR avionics applications, addressing critical gaps in technical specifications, safety protocols, and standardization for SDAM requirements. Through strategic integration of RF and power components, the resulting architecture was not only modular and configurable but also finely tuned to diverse operational demands across various SDAMs. Significant enhancements were achieved in SWaP-C parameters, demonstrating the design’s capacity for scalability and improved efficiency.

Validation efforts focused on key performance metrics, including signal power, receiver gain, and linearity. The experimental framework, utilizing the operational frequencies of avionic radios such as DME, ADS-B, and TMS, confirmed a consistent gain of 18 dB across all devices tested, with reliable performance from −90 dBm to −20 dBm input signal levels.

Moreover, this project pioneered a sophisticated method for managing multiple RFFE inputs and outputs, facilitating the operation of high-power transmitters and ultra-sensitive receivers without sacrificing safety or performance. This methodology ensures comprehensive isolation and interference management, supporting a robust performance spectrum suitable for complex avionic systems. Collectively, these advancements underscore the significant potential of this agile RFFE system to enhance the reliability and functionality of avionics communications, setting a new standard for future developments in the field.

## Figures and Tables

**Figure 1 sensors-24-05963-f001:**
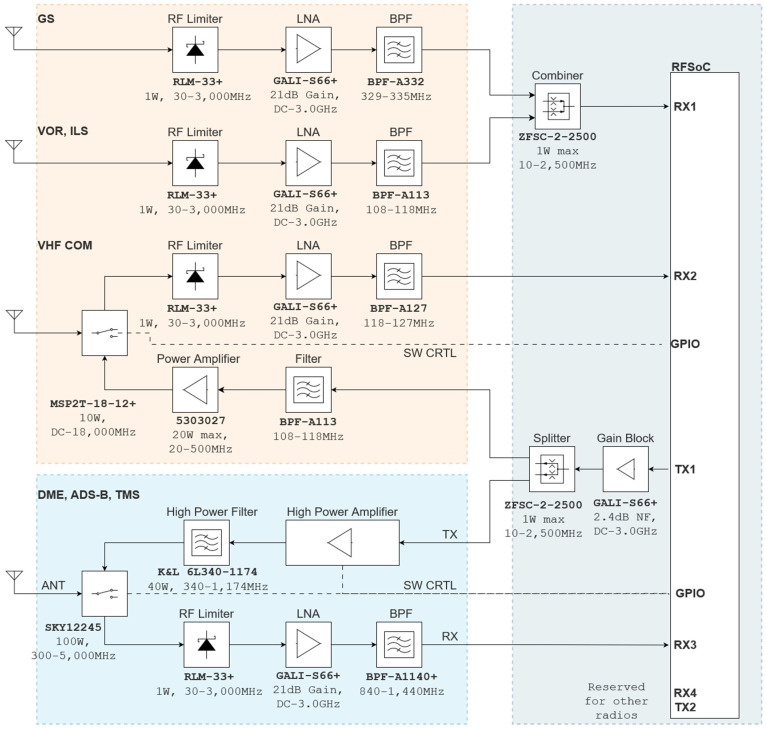
Proposed architecture block diagram showing the RF paths.

**Figure 2 sensors-24-05963-f002:**
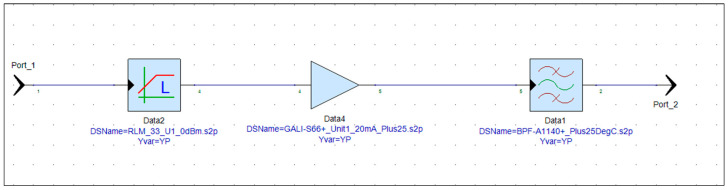
VOR receiver S-parameter block diagram.

**Figure 3 sensors-24-05963-f003:**
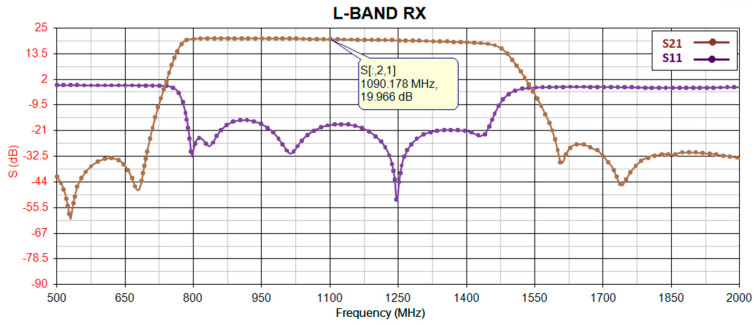
DME, ADS-B, and TMS gain (S21) and input-matching (S11) simulation.

**Figure 4 sensors-24-05963-f004:**
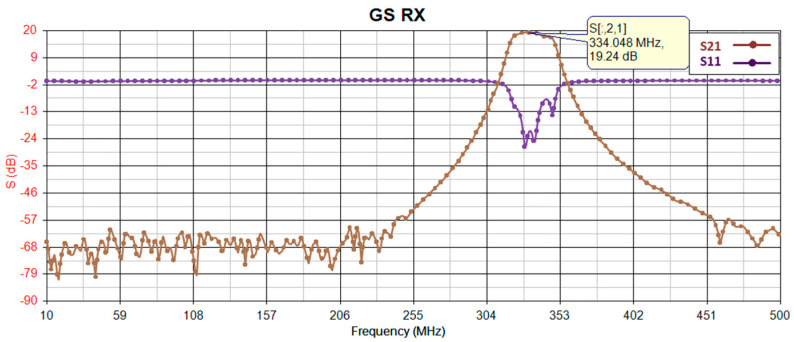
Glide Slope receiver gain (S21) and input-matching (S11) simulation.

**Figure 5 sensors-24-05963-f005:**
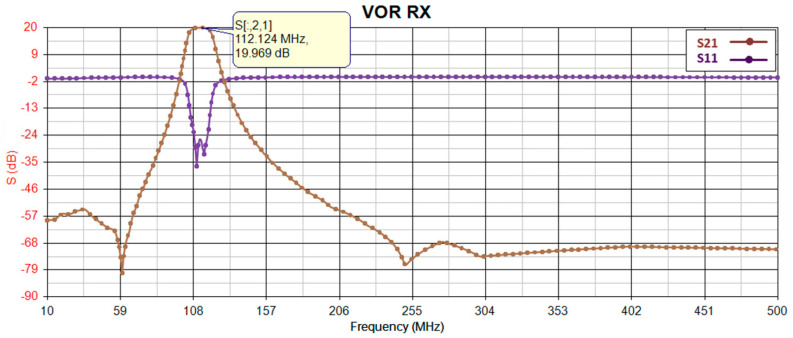
VOR/ILS LOC gain (S21) and input-matching (S11) simulation.

**Figure 6 sensors-24-05963-f006:**
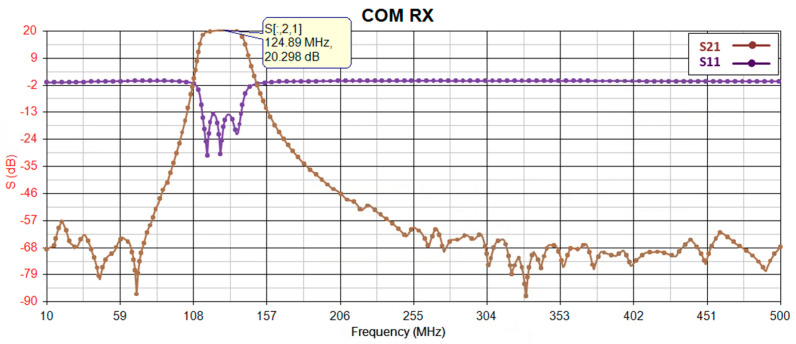
COM gain (S21) and input-matching (S11) simulation.

**Figure 7 sensors-24-05963-f007:**
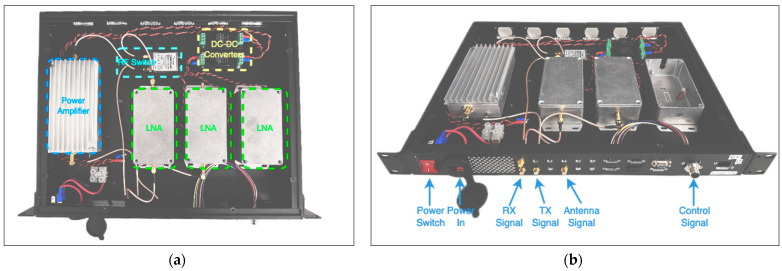
(**a**) VHF Box. Picture of the VHF rack unit’s design and assembly. (**b**) VHF rack unit’s ports’ description.

**Figure 8 sensors-24-05963-f008:**
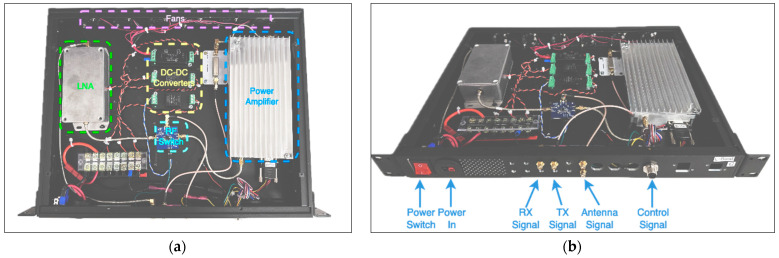
(**a**) L-band box arrangement. L-band rack unit’s design and assembly. (**b**) L-band rack unit’s ports’ description (**b**).

**Figure 9 sensors-24-05963-f009:**
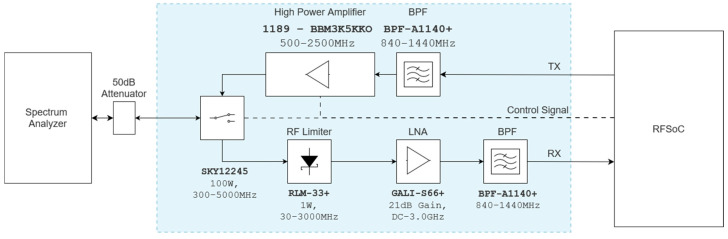
Diagram of the L-band module connected to a spectrum analyzer.

**Figure 10 sensors-24-05963-f010:**
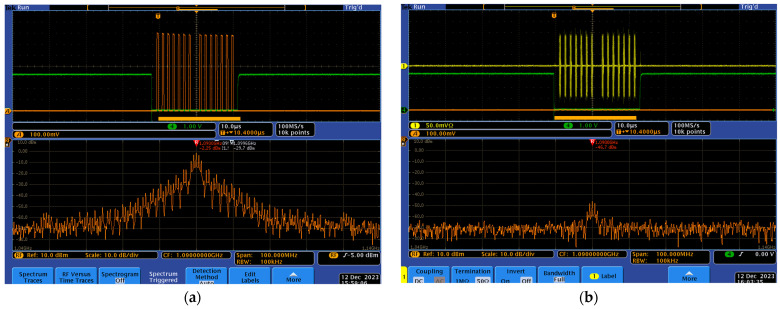
(**a**) Frequency and time-domain signals of ADS-B at the transmitter antenna (50 dB attenuation). (**b**) Frequency- and time-domain signal of ADS-B at the receiver (50 dB attenuation) (**b**).

**Figure 11 sensors-24-05963-f011:**
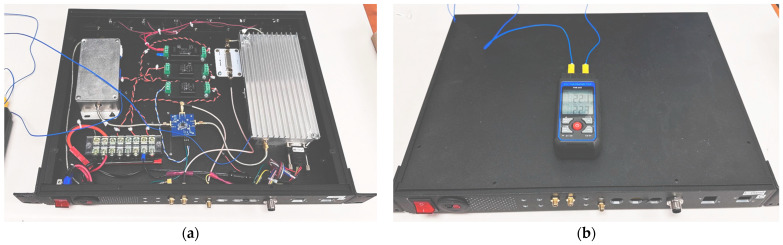
(**a**) Thermocouple probes’ placement in the L-band radio rack. (**b**) Final temperature measurement-test setup before test begins at room temperature.

**Figure 12 sensors-24-05963-f012:**
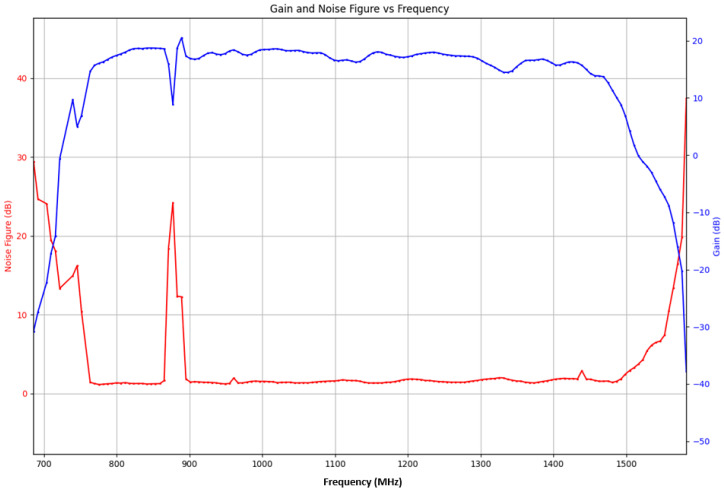
Noise figure and gain-measurement result of the L-band receiver block.

**Figure 13 sensors-24-05963-f013:**
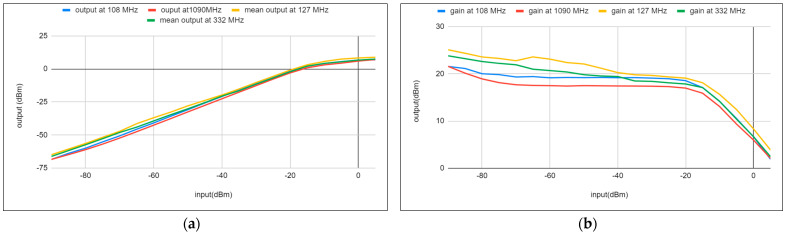
(**a**) Receivers block output power gain. (**b**) Linearity of all the receive paths vs. input signal power.

**Figure 14 sensors-24-05963-f014:**
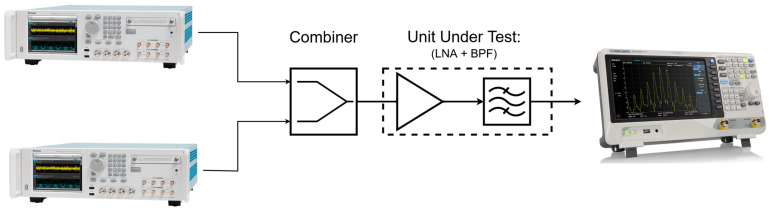
Two-tone measurement setup.

**Figure 15 sensors-24-05963-f015:**
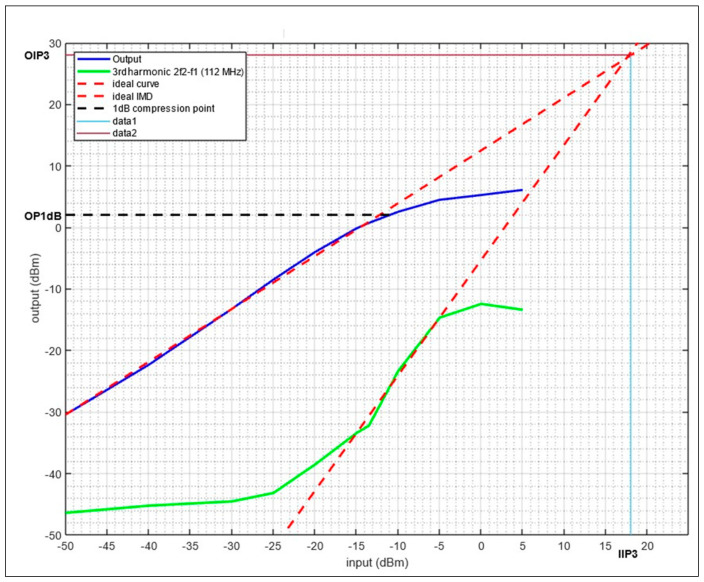
OP1dB and IIP3 measurement for the LNA with a two-tone input at 108 MHz/110 MHz.

**Figure 16 sensors-24-05963-f016:**
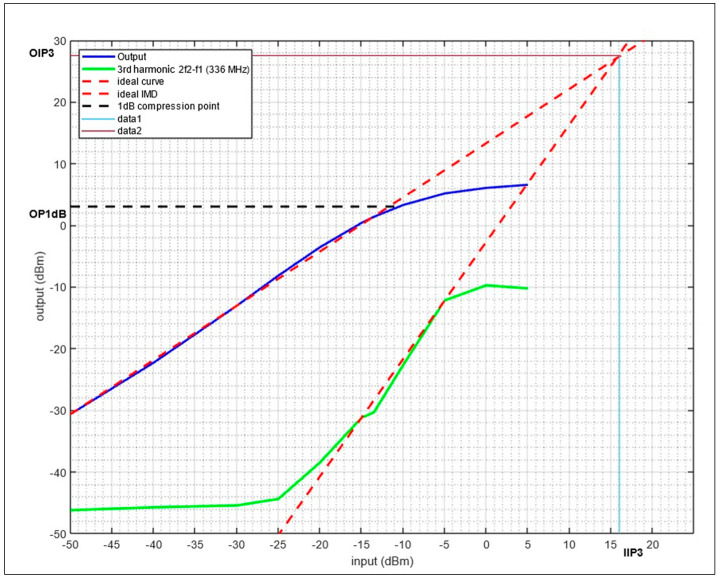
OP1dB and IIP3 measurement for the LNA with a two-tone input at 330 MHz/332 MHz.

**Figure 17 sensors-24-05963-f017:**
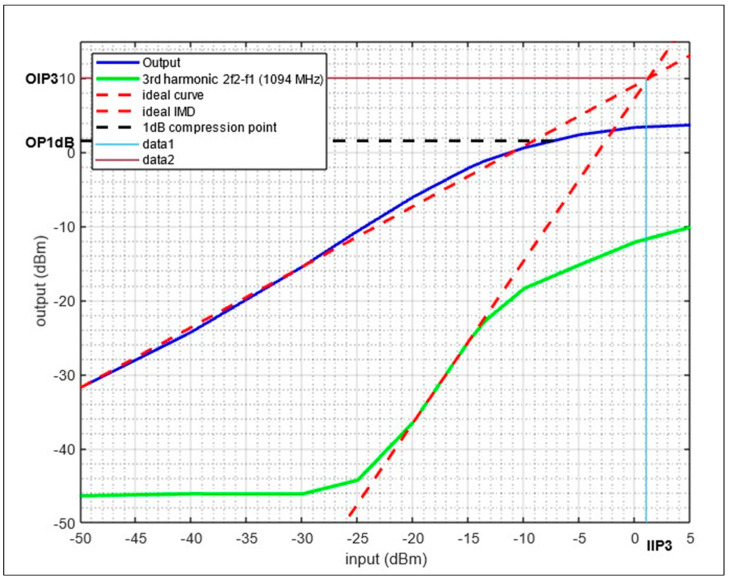
OP1dB and IP3 measurement for LNA with a two-tone input at 1090 MHz/1092 MHz.

**Figure 18 sensors-24-05963-f018:**
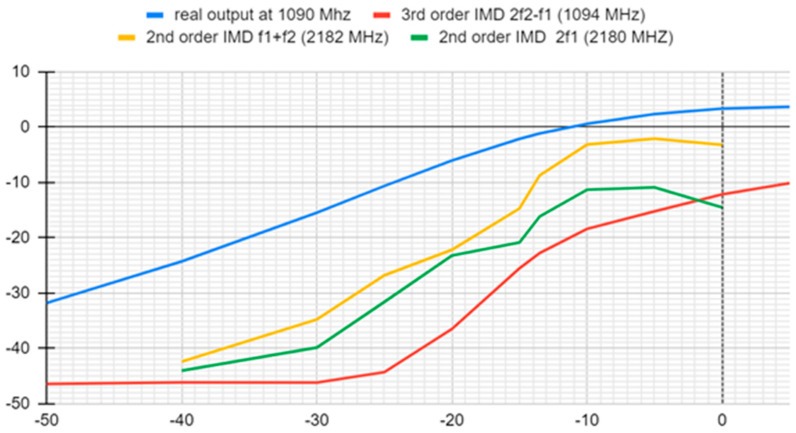
First-, second-, and third-order IMD comparison at a 1090 MHz frequency.

**Table 1 sensors-24-05963-t001:** Different avionics radio specifications based on DO standard.

Radio	Aviation Standard	Center Freq. (MHz)	BW (MHz)	Modulation	RX Sensitivity (dBm)	TX Power (dBm)
VOR	DO-196	108	0.5	AM	−120	0
ILS LOC	DO-195	108	0.5	AM	−120	0
ILS GS	DO-172	332	0.5	AM	−120	0
COM	DO-163	118	5	AM	−110	40
ADS-B	DO-242	1090	1	Model S	−90	50
DME	DO-189	1090	1	Pulse	−100	50
TMS	DO-181D	1090	1	AM	−110	50

**Table 2 sensors-24-05963-t002:** RF budgeting of the receiver path.

	Antenna	RF Limiter	LNA	BPF	ADC
Loss/gain (dB)		0.2	20	2	-
Min output power (dBm)	−75	-	−55	−57	-
IP1dB (dBm)	-	-	3.3	1.3	-
Max output power (dBm)	-	12	12.2	10.2	-
Absolute max input (dBm)		30	20	26.99	14.6

**Table 3 sensors-24-05963-t003:** RF budgeting for the TX-to-RX leak.

	Receiver Block	RF Limiter	LNA	BPF	ADC Input
Loss/gain (dB)	-	0.2	20	2	-
Input					
(dBm)	3.3	3.3	3.1	12.1	12.1

**Table 4 sensors-24-05963-t004:** Measured temperatures of two probes in a two-hour test.

Operation Time (Minutes)	LNA Body Temperature (Degrees)	PA Body Temperature (Degrees)
0	25.6	25.4
10	32.2	26.8
20	38.3	28.9
30	42.6	31.2
60	45.7	33.1
90	46.6	34.2
120	47.1	35.5
150	47.2	35.8

**Table 5 sensors-24-05963-t005:** Gain/insertion loss of components used in each receiver block.

	RF Limiter Loss	LNA Gain	BPF Loss	Overall Gain
VOR, ILS LOC (dB)	0.23	21.6	2.10	18.27
COM (dB)	0.23	21.8	2.10	19.47
ILS GS (dB)	0.23	21.2	3.01	17.96
TMS, DME, ADS-B (dB)	0.21	20.3	0.99	17.21

## Data Availability

Data are contained within the article.

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
