# Peer review of "An Adaptive RF Front-End Architecture for Multi-Band SDR in Avionics"

_sensors, 2024, doi:10.3390/s24185963_

Round 1
Reviewer 1 Report
Comments and Suggestions for Authors
The article presents a multi-band RF transceiver architecture to deal with communications in avionics. Agility and flexibility are reached through the use of RFSoC-based SDR HW. The authors claim their approach both addresses RF peformance and SWAP-C requirements.
To my understanding this is not a research work, but an application-oriented RF engineering work based on general practices which may have some practical interest to some professionals of the RF avionics community. The authors propose some multi-band/comms RF Transceiver architecture based on COTS RF modules. This approach makes hard to give higher value to the engineering effort since there is no novel RF design and I cannot be certain on the architecture novelty. There is neither a quantitative comparison with other SotA architectures/designs or even commercial solutions. Also there are several claims that can hardly be demonstrated and some information is missing. Some further clarification may be required regarding the following aspects:
- No standard requirements have been described/set for the different communication links or RF modules. For example, there is neither an E2E communication link budget nor a RF system budget analysis/simulation/measurement, or a minimum reference to standard-supported requirements helping to show their approach is optimal in terms of performance. The article lacks some technical background and systematical approach.
- I am probably lacking some further context information. For example, how are the antennas physically distributed, are the antennas sufficiently spaced so no undesired couplings exist, which are the losses in between the T/R switches and the antenna? In Figure 1, what prevents the GS & VOR and the ILS to use common RX RF chain with broader BPF before A/D conv. & digital filtering at the RFSoC?
- In Figures 2-3-4-5-6, how can the noise figure (NF) be so low? The NF of the LNA seems not to have been accounted in the calculation/simulation. Less important, but worth mentioning, the RF cables used for interconnection between RF modules will also show some small losses that need to be accounted in between the antenna and the RF limiters/switches, and between those and the LNAs. Why the authors have not provided a measurement of the NF for their RF TRX rack units to validate their simulation?
- While it is clear that the PA is switched off when the RX is operating, it seems the RX LNA is switched on when the TX PA is operating. If 3-4 dBm are coupled from the TX output to the Rx input can this have any effect on the LNA due to having to handle both high and low power values (e.g. trapping effects)? Is this really a 'negligible effect' as claimed by the authorss (L338). Following the RX chain, on the other hand, it seems the Rx ADC is already protected through a power detector+variable attenuator).
- VHF Radio Design: The authors refer ot 28->12V DC-DC converters but do not comment what else is done to bias the RX LNAs at lower voltage.This comment can also apply to the L-band Radio Design also having the same LNAs.
- L-band Radio Design: 'Additional elements include a preamplifier for signal conditioning and a variable attenuator controlled by a power detector to adjust the RX pin signal levels on the RFSoc' -> There is no visibility of these modules at the TRX block diagrams or at the pictures of the RF rack units. There is neither visibility on the HPA module at the L-band TX delivering 50W. Which are its COTS building blocks? Will the ON/OFF operation of the PA provoke undesired trapping or long term memory effects?
- 4.5 Heat Management: I am probably missing some information on how frequently the transmissions are produced to understand how effective the PA on/off switching will be to mitigate thermal issues. Regarding the measurements over room temperature, why haven't the authors done the measurements at higher operating temperatures? The thermocouples seem to be attached to the RF module enclosures/heat sinks. Haven't the authors conducted either a static or a dynamic thermal analysis/simulation to estimate the active devices junction temperatures accounting for the worst case thermal conditions?
- RF performance: The authors claim that they improve 18 dB the sensitivity of the receiver but this seems not a very relevant merit when it is just the direct effect of adding an LNA and take care of having low loss before the LNA. They should compare the improvement with respect to using other RF transceiver approaches, instead. Regarding the OIP3 how can the OIP3 be about 10 dB higher to what is specified at the datasheet? Why rather than making projections, haven't the authors evaluated the OIP3 by visualizing the two-tone testing measurement at the spectrum analyzer as it seems doable? Finally, I missed some measurement showing the TX path linearity, harmonics, spurious emission, etc.
- Discussion: The authors leverage on COTS modules. While they try to share parts, this approach cannot be optimally in terms of SWAP when compared to a customly tailored RF design. Regarding reliability and resilience, some further proof should be provided as stated in the comments at 4.5 Heat Management and, generally speaking, better providing an holistic analysis comparing the traditional approach with that proposed by the authors.
ADDITIONAL COMMENTS:
- Text at (L62) should be moved to appear below Table 1.
- Re-format Section 3. System architecture (L161) title.
- Review ALL Figure reference insertions since there are numerous errors like i) missing a space between the Figure reference and the word following the reference, ii) wrong numbering, iii) missing dots at sentence finished with figure reference.
- Review section numbering at 2.3 (L204, should be 3.3).
- Skyworth (L335)
Author Response
Thank you for your detailed review and insightful feedback. We acknowledge that this work focuses more on application-oriented RF architecture design. Given the legacy nature of aviation transmit and receive technology, it is crucial for any new system, including advanced SDR, to maintain compatibility with existing systems. This necessitated a straightforward, flexible, and precise RF front-end design.
The proposed architecture has undergone extensive testing, documentation review, and refinement to achieve the current results. It represents a significant effort to ensure practical applicability and reliability within the RF avionics community in many aspects, notably due to its support of several radios. While the architecture may appear similar in the block diagrams, the detailed design and selection of each module and interconnections between blocks provide the flexibility that distinguishes this work.
We will address in the following each point separately with supporting evidence to provide further clarification in this document

Reviewer 2 Report
Comments and Suggestions for Authors
In this article, the authors introduce a reconfigurable and agile RF front-end architecture to enhance the performance of software-defined radios (SDRs) by adjusting signal requirements, frequencies, and protocols. Integration of flexible RF front-end in SDRs, with design of essential components such as receivers, transmitters, RF switches, combiners, splitters, and their corresponding RF pathways are presented. The presented research work is interesting however, I do not recommend its acceptance in its present form. I have following concerns.
1) Presentation of the work in the manuscript need to improve for example, see the text in between Table 1 and Table caption as well as there is not any reference in Section 1 (Introduction). The authors contribution in the manuscript is not clearly presented which need to clarify in the bullet form.
2) How does the adaptive RF front end architecture develop? How is it novel? Is any algorithm/protocol developed? It needs to clarify in detail.
3) Fig. 7 to Fig. 10 could be presented in one figure as (a), (b), (c) and (d) with clarity of explanation particularly its adaptive nature.
4) I did not find the detail discussion about adaptive receiver features such as dynamic range etc..
Comments on the Quality of English Language
Minor editing is required.
Author Response
Thank you for your detailed review and insightful feedback. We have carefully considered your valuable comments and revised the paper accordingly. In the following document, we outline how each comment has been addressed.

Round 2
Reviewer 1 Report
Comments and Suggestions for Authors
The authors have enhanced the quality and technical soundness of the manuscript while reasonably addressing the comments made during the first review. For this reason I think the manuscript can be accepted in its current form.
Reviewer 2 Report
Comments and Suggestions for Authors
The authors have incorporated the issues raised by reviewers in the revised manuscript. So, I recommend the acceptance of the manuscript.